# Hepatitis E Outbreak in the Central Part of Italy Sustained by Multiple HEV Genotype 3 Strains, June–December 2019

**DOI:** 10.3390/v13061159

**Published:** 2021-06-17

**Authors:** Anna Rosa Garbuglia, Roberto Bruni, Umbertina Villano, Francesco Vairo, Daniele Lapa, Elisabetta Madonna, Giovanna Picchi, Barbara Binda, Rinalda Mariani, Francesca De Paulis, Stefania D’Amato, Alessandro Grimaldi, Paola Scognamiglio, Maria Rosaria Capobianchi, Anna Rita Ciccaglione

**Affiliations:** 1National Institute for Infectious Diseases “L. Spallanzani” IRCCS, 00149 Rome, Italy; annarosa.garbuglia@inmi.it (A.R.G.); francesco.vairo@inmi.it (F.V.); daniele.lapa@inmi.it (D.L.); paola.scognamiglio@inmi.it (P.S.); maria.capobianchi@inmi.it (M.R.C.); 2Viral Hepatitis and Oncovirus and Retrovirus Diseases Unit, Department of Infectious Diseases, Istituto Superiore di Sanità, 00161 Rome, Italy; umbertina.villano@iss.it (U.V.); elisabetta.madonna@iss.it (E.M.); annarita.ciccaglione@iss.it (A.R.C.); 3Department of Infectious Diseases, S. Salvatore Hospital, 67100 L’Aquila, Italy; giovanna.picchi.inf@gmail.com (G.P.); a.grimaldi62@gmail.com (A.G.); 4General and Transplant Surgery Unit, ASL 1 Abruzzo, S. Salvatore Hospital, 67100 L’Aquila, Italy; bindabarbara@gmail.com; 5Infectious Diseases Unit, Department of Medicine, SS Filippo and Nicola Hospital, 67051 L’Aquila, Italy; rmariani@asl1abruzzo.it; 6Veterinary Service for the Hygiene of Foods of Animal Origin and Their Derivatives, Department of Prevention, ASL1 Avezzano Sulmona L’Aquila, 67100 L’Aquila, Italy; fdepaulis@asl1abruzzo.it; 7Ministry of Health, 00144 Rome, Italy; stefania.damato@sanita.it

**Keywords:** HEV, outbreak, genotype, sub-genotype

## Abstract

In European countries, autochthonous acute hepatitis E cases are caused by Hepatitis E Virus (HEV) genotype 3 and are usually observed as sporadic cases. In mid/late September 2019, a hepatitis E outbreak caused by HEV genotype 3 was recognized by detection of identical/highly similar HEV sequences in some hepatitis E cases from two Italian regions, Abruzzo and Lazio, with most cases from this latter region showing a link with Abruzzo. Overall, 47 cases of HEV infection were finally observed with onsets from 8 June 2019 to 6 December 2019; they represent a marked increase as compared with just a few cases in the same period of time in the past years and in the same areas. HEV sequencing was successful in 35 cases. The phylogenetic analysis of the viral sequences showed 30 of them grouped in three distinct molecular clusters, termed A, B, and C: strains in cluster A and B were of subtype 3e and strains in cluster C were of subtype 3f. No strains detected in Abruzzo in the past years clustered with the strains involved in the present outbreak. The outbreak curve showed partially overlapped temporal distribution of the three clusters. Analysis of collected epidemiological data identified pork products as the most likely source of the outbreak. Overall, the findings suggest that the outbreak might have been caused by newly and almost simultaneously introduced strains not previously circulating in this area, which are possibly harbored by pork products or live animals imported from outside Abruzzo. This possibility deserves further studies in this area in order to monitor the circulation of HEV in human cases as well as in pigs and wild boars.

## 1. Introduction

Hepatitis E Virus (HEV), responsible for human hepatitis E, is classified in the *Hepeviridae* family, which includes viruses infecting either humans or animal species or both [1]. The *Hepeviridae* family includes two genera: *Orthohepevirus* and *Piscihepevirus*. Members of the *Orthohepevirus* genus are sub-classified into four species: *Orthohepevirus A* to *D* [1].

Most HEVs detected in humans have been so far classified into the *Orthohepevirus A* species until a recent report that demonstrated frequent transmission to humans of rat HEV, which is a member of the *Orthohepevirus C* species [2]. It remains to be investigated how widespread rat HEV infection is in humans worldwide [3].

According to HEV genome sequences, all *Orthohepevirus A* viruses detected in humans have been classified into four genotypes (HEV-1 to HEV-4), with the unique exception of a genotype 7 virus (HEV-7) reported in a single case [4]. No further HEV-7 related cases have been so far documented and so it is likely this latter genotype, for which its natural host is camel, could be transmitted under exceptional circumstances [5].

HEV-1 and HEV-2 exclusively infect humans, displays fecal–oral transmission through contaminated water, and are responsible for both sporadic cases and waterborne epidemics. Acute liver failure caused by HEV-1 was reported in elderly patients and in a leukemia patient [6,7]. They are endemic in developing countries; HEV-1 circulates especially in Central and South Asia (in particular India) and North Africa; HEV-2 mainly circulates in Central America, Mexico, and West Africa [8,9]. In developed countries, these genotypes can be detected in sporadic cases and are related to travel in areas in which HEV-1 and HEV-2 are endemic.

In contrast, HEV-3 and HEV-4 have been detected both in humans and in animal species (HEV-3 in pig, wild boar, deer, mongoose, and rabbits; HEV-4 in pig). HEV-3 and HEV-4 are mainly transmitted by the consumption of products from infected animals, especially undercooked meat (including pork liver and liver containing pork products) [10]. Potential HEV reservoir might also be represented by cows, sheep and goats; although, to date, HEV transmission to humans from these species has not been documented, while HEV detection in milk samples from these species was reported in China, Turkey, Egypt, and Europe, with potential risk for zoonotic transmission [11,12,13,14,15]. HEV was also detected in Italy in fecal samples from goat and fecal and serum samples from sheep [16,17]. While circulation of HEV-4 is mostly limited to East-Asia, HEV 3 is diffused worldwide and is considered an emerging pathogen in western countries, including Europe [8,9]. According to a meta-analysis published in 2016, the prevalence of anti-HEV IgG in the general population and in blood donors in European countries is highly heterogeneous, ranging from 0.6% to 52.5% and implying very high circulation in some areas [18]. In Italy, a nationwide study in blood donors from the 21 Regions/Autonomous Provinces showed heterogeneous mean prevalence rates, ranging between 2.2% and 22.8%; extremely high rates, which are those exceeding 30%, were observed in some areas of Sardinia and Abruzzo [19].

Most HEV infections in Europe are detected as sporadic cases; previously described European outbreaks caused by HEV-3 were linked either to familial consumption of wild-boar meat or to undercooked pig liver-based stuffing at a wedding meal [20,21].

The present paper describes a HEV-3 outbreak sustained by subtype 3e and 3f strains, which occurred in Central Italy in June 2019 to November 2019. Most cases were linked to the consumption of pork products at restaurants or festivals in the Abruzzo region; the outbreak strains formed three clusters independent from the strains detected in previous years in sporadic cases from this area. An effective and coordinated “One health” collaboration brought the recognition and evaluation of the outbreak.

## 2. Materials and Methods

### 2.1. Case Definition

Due to the different clinical course in immunocompetent and in immunosuppressed transplanted individuals, the following case definitions were used:Confirmed cases:
-Any acute hepatitis case with onset of symptoms after 8 June 2019, with a serum sample positive for anti-HEV IgM and/or HEV RNA; a viral sequence in the ORF2 region identical to or clustering with a bootstrap value > 70 in a phylogenetic tree; with one of the following three HEV reference sequences: Accession number MN497623, MN537879, or MN737483;-Any transplanted patient or blood donor with a serum sample positive for HEV RNA sampled after 8 June 2019 and following a previous negative test and a viral sequence in the ORF2 region that is identical or clustering with a bootstrap value > 70 in a phylogenetic tree; with one of the following three HEV reference sequences: Accession number MN497623, MN537879, and MN737483.Possible cases:
-Any acute hepatitis case with onset of symptoms after 8 June 2019 with a serum sample positive for anti-HEV IgM and/or HEV RNA (but no available viral sequence);-Any transplanted patient or blood donor with a HEV RNA positive serum sample collected after 8 June 2019, following a previous negative test (but no available viral sequence).Non outbreak cases:
-Any acute hepatitis case with onset of symptoms after 8 June 2019 with a serum sample positive for anti-HEV IgM and/or HEV RNA and a viral sequence in the ORF2 region located in a phylogenetic tree in a different branch vs. the following three HEV reference sequences: Accession number MN497623, MN537879, and MN737483 and with no clustering with other sequences collected after 8 June 2019;-Any transplanted patient or blood donor with a HEV RNA positive serum sample collected after 8 June 2019 (following a previous negative test) and a viral sequence in the ORF2 region located in a phylogenetic tree in a different branch vs. the following three HEV reference sequences: Accession number MN497623, MN537879, and MN737483 and with no clustering with other sequences collected after 8 June 2019.

### 2.2. Epidemiological Investigation

Demographic and epidemiological data were collected upon interview of each person with a standard structured questionnaire under the public health investigation activities. Collected data included gender, age, residence, level of education, job, as well as information about travel (abroad or in Italy outside the place of residence); consumption of raw/undercooked meat, sausages, or liver sausages from swine or wild boar; consumption of raw/undercooked meat from deer; consumption of shellfish; direct contact with pig or wild boar; small farming of pigs for self-consumption; exposure to pig manure, hunting, and having drunk well water.

### 2.3. HEV RNA and Anti-HEV IgM/anti-HEV IgG Detection

HEV RNA was detected by the RealStar HEV RT-PCR kit (Altona Diagnostics, Hamburg, Germany) according to the manufacturer’s instructions. Template was cDNA from one half RNA extracted from 200 μL plasma or serum samples.

Anti-HEV IgM and IgG were tested by the Wantai HEV-IgM ELISA and Wantai HEV-IgG ELISA (Beijing WANTAI Biological Pharmacy Enterprise Co. Ltd., Beijing, China) in patients from Abruzzo and by DIA.PRO HEV IgM ELISA and HEV IgG ELISA (DIA.PRO, Milan, Italy) in patients from Lazio.

### 2.4. Sequencing and Sequence Analysis

Viral characterization was carried out at ISS and INMI by RT-PCR amplification and sequencing of a fragment of the viral genome (ORF2 region).

The ISS method results in a 493 nt sequence. Details of the method, which includes primer sequences made available by RIVM to European laboratories participating in the HEVnet database, were reported previously [22,23]. Briefly, viral RNA was extracted from 200 μL serum by using the QIAmp MinElute Virus Spin kit (Qiagen, Hilden, Germany). One seventh extracted RNA (5 μL) was reverse transcribed by the SuperScript III First-Strand Synthesis System for RT-PCR (Invitrogen, a brand of Thermo Fisher Scientific, Waltham, MA, USA) with random hexamers. Nested PCR primer sequences were previously published [22]. For the first PCR step, PCR cycling conditions were as follows: 35 cycles of 95 °C 30 s; 42 °C 30 s; 60 °C 45 s. For the second (nested) PCR, PCR cycling conditions were as follows: initial denaturation 95 °C 6 min and then 40 cycles of 9 5°C 30 s followed by 60 °C 20 s and 72 °C 15 s.

The INMI method results in a 412 nt sequence. Details of the method were reported previously [24,25]. Briefly, viral RNA was extracted from 400 µL of serum or plasma using the QIASYMPHONY automated instrument (QIAGEN, Hilden, Germany). cDNA was retro-transcribed using the SUPERSCRIPT IV reverse transcriptase first round system (Thermofisher Scientific, Paisley, United Kingdom) with random hexamer primers according to the manufacturer’s instructions. A nested PCR technique was used to amplify DNA fragments within the ORF2 (positions 5953—6363 respect to the E116-YKH98C strain, AB369687). The first round PCR was performed using 10 µL cDNA in a total mixture of 50 µL containing 10× buffer, 2.5 µL MgCl2 (20 mM MgCl_2_ stock solution), 10 mM dNTP, 2.5 IU TaqGold polymerase (Applied Biosystem, Foster City, CA, USA), and 0.5 µL each of forward and reverse outer primers (50 µM). The same conditions were employed for the second round PCR except 5 µL of the first round product was used as a template. The following primers were used in the first round: HE44- 5′CAAGGHTGGCGYTCKGTTGAGAC3′ and HE40- 5′CCCTTRTCCTGCTGAGCRTTCTC3′. Primers used in the second round were HE110-5′GYTCKGTTGAGACCTCYGGGT3′ and HE41-5′ TTMACWGTCRGCTCGCCATTGGC3′ [24].

An overlapping 370 nt region, shared by the ISS and INMI sequences, was used for building the phylogenetic tree. Phylogenetic and distance analyses were carried out by MEGA [26,27] as previously described [23]. The sequences of the present study were deposited in GenBank (Accession number MN497623, MN537879, MN737482, MN737483, MN737484, LN681545 and MZ274228 to MZ274271).

## 3. Results

### 3.1. Outbreak Detection

The outbreak was first recognized in mid/late September by the comparison of viral sequences obtained from anti-HEV IgM positive acute hepatitis patients with onset between 8 June and 21 September 2019. The sequences were obtained in the frame of routine virological characterization for surveillance purposes by the Regional Reference Laboratory (RRL) established at the National Institute for Infectious Diseases, “L. Spallanzani” (INMI) for cases from the Lazio region, and by the National Reference Laboratory (NRL) at Istituto Superiore di Sanità (ISS) for cases from the Abruzzo region.

Specifically, HEV sequences from eight patients (onset 19 July to 11 September) in the Lazio region were recognized in mid-September to be identical; an additional case with the same sequence was notified later (onset 20 September) and raised the number of patients with identical HEV sequence to nine. Two further cases were also observed: one case (onset: 28 June) showed a HEV sequence highly related, but not identical, to those from the above nine cases while no sequence could be obtained from the other. Overall, eleven hepatitis E cases were observed between 28 June and 20 September in Lazio: analysis of routine surveillance data showed that they represented a five-fold increase in hepatitis E cases compared with the same period of the previous year (data not shown). The Regional Service for Epidemiology, Surveillance, and Control of Infectious Diseases (SERESMI) of Lazio was promptly notified by the RRL to begin analysis of available epidemiological data from notified cases as well as to start further epidemiological investigations. Eight of the eleven patients indicated a stay in Abruzzo in from June to August and with local consumption of swine meat.

Roughly in the same period of time (late September), the NRL received serum samples from thirteen patients with acute hepatitis: twelve anti-HEV IgM positive cases and one suspected nonA-nonC anti-HEV IgM negative case (onset 8 June to 21 September). In the past years just a few (1–2) or zero hepatitis E cases had been reported from Abruzzo in June-September. The samples were sent from three hospitals in the Abruzzo region, that are actively involved in the national surveillance for HEV and routinely sends serum samples from anti-HEV IgM positive cases with acute hepatitis to the NRL. The clinicians highlighted that some of these cases showed unusual presentation with more severe symptoms than patients observed in the past years, high ALT levels, and cholestasis and also in young people without any comorbidities. In addition to the hospitalized cases, three asymptomatic HEV RNA positive acute infections were also identified during late June and early July among solid organ transplanted individuals attending the Regional Transplant Centre of Abruzzo and Molise. They were detected in the frame of a collaborative active surveillance project between the ISS and the Centre. The project was established in 2019 to screen transplanted individuals for HEV infection on a periodical basis. In fact, the high HEV circulation in Abruzzo, suggested by seroprevalence data [19], poses transplanted patients at special risk: it is well known that in these patients acute HEV infection may evolve into chronic and is ultimately responsible for accelerated development of cirrhosis [28]. Analysis of HEV sequences obtained from 11 of the 16 cases from Abruzzo showed two distinct clusters, subtype 3e, as well as two unique unrelated sequences, subtype 3f.

The overall analysis of HEV sequences from Lazio and Abruzzo clearly showed two related but well distinct subtype 3e clusters, from here onwards named “Cluster A” and “Cluster B” (see the phylogenetic analysis below), which were not yet geographically grouped: members of both clusters were observed from both Lazio and Abruzzo.

Further hepatitis E cases occurred thereafter (late September to early December) in both Lazio (*n* = 9) and Abruzzo (*n* = 10); HEV sequences could be obtained from 13 of them. Phylogenetic analysis showed an independent subtype 3f cluster including 11 heterogeneous but highly related sequences that are well distinct from other 3f strains: this group was named “Cluster C”. As to the remaining two sequences, one of them was identical to the 3e strains of Cluster B while the other appeared as a unique 3f sequence unrelated to the members of Cluster C.

Overall, 47 cases of HEV infection were finally observed between 8 June and 6 December 2019. They included 44 hospitalized acute hepatitis patients and three asymptomatic transplanted individuals. The median age was 63 years (range: 28–85 years) and the majority (77.3%) were males. The median viral load for 36 evaluable cases was 1.5 × 10^4^ HEV-RNA copies/mL (range: 1 × 10^1^–9 × 10^5^ copies/mL). The liver function test median values for 32 evaluable cases were alanine aminotransferase (ALT), 1209 U/L (range: 33–6400); aspartate aminotransferase (AST), 921 U/L (range: 21–4712); total bilirubin, 5 (range: 1–23 mg/dL). All three transplanted individuals had undergone kidney transplantation from 4 to 9 years earlier due to kidney cancer or polycystic kidney disease and were under immunosuppressive therapy maintenance with tacrolimus.

With the exception of two patients (one transplanted patient testing negative for both anti-HEV IgM and anti-HEV IgG and an acute hepatitis patient testing anti-HEV IgM negative and anti-HEV IgG positive), all the others tested positive for both anti-HEV IgM and anti-HEV IgG.

### 3.2. Virological Characterization

The viruses from 35 of 47 cases occurred in Abruzzo and Lazio (onset between 8 June 2019 and 6 December 2019) could be molecularly characterized by PCR amplification and sequencing.

Figure 1 shows the results of phylogenetic analysis of the HEV sequences from the cases identified in the present study (red circles) in the background of sequences from cases that occurred in Abruzzo and Latium in the first half of 2019 (light gray circles) and in the past years in Abruzzo (dark gray circles), as well as of reference sequences; the sequence of one outbreak case was excluded from phylogenetic analysis because it was much shorter than the other ones (see the legend of Figure 1).

Three clusters (termed A, B, and C) can be recognized: cluster A and cluster B were caused by 3e strains and cluster C by 3f strains. In cluster B, all sequences were identical; in cluster A and C, intra-cluster heterogeneities were observed (0–3 nt differences among strains of cluster A; 0–6 nt differences among strains of cluster C). The mean intragroup distances in Cluster A (*n* = 4) and C (*n* = 11) were 0.0045 and 0.0057 substitutions/site, respectively. One sequence (ISS_ID 331/2019) had molecular features intermediate between cluster A and B.

The phylogenetic tree clearly shows that the 3e and 3f strains from Clusters A, B, and C are poorly related to the strains detected in Abruzzo in the past years. Actually, in the past years, exclusively 3f and 3c strains had been detected in the area of Abruzzo involved in the present outbreak but no 3e strains were detected: these latter had been observed exclusively in the coastal area and are poorly related to the strains of the present outbreak (ISS_ID259/2018, ISS_ID261/2018, ISS_ID60/2008, and ISS_ID145/2015). Sequence comparison with HEV strains deposited in the HEVnet European database showed high similarity with strains circulating in East Europe. Overall, molecular data suggests that the outbreak might have been caused by the newly introduced viruses that were not previously circulating in the areas involved in the outbreak.

Three cases observed in June to October (one transplanted person and two acute hepatitis subjects: ISS-Transpl ID255, ISS_ID321/2019, and ISS_ID340/2019) showed 3f viral sequences unrelated to each other that were also unrelated to the 3f outbreak strains in Cluster C; these cases showed no epidemiological link. Thus, they most likely represent the sporadic cases unrelated to the outbreak that are caused by endemic strains circulating in Abruzzo.

### 3.3. Outbreak Curve

In the period from 1 June to 6 December, 47 cases of HEV infection were observed in Abruzzo and Latium. Figure 2 shows the outbreak curve with cases classified according to virological characterization and case definitions.

The three molecular clusters (Cluster A, B, and C) partially showed overlapped temporal distribution: three Cluster A cases occurred in June and a fourth case in September, 15 Cluster B cases occurred from July to October, and 11 Cluster C cases occurred from September to December.

Twelve “possible outbreak cases” were observed from June to November (i.e., anti-HEV IgM positive cases without HEV RNA sequencing and classification due to low or undetectable viral RNA concentration). In the same period, four additional cases proved to be “sporadic” cases: the underlying HEV strains were unique and unrelated to any member of the three outbreak clusters (subtypes: three 3f and one 3c. See the red circles placed on different branches of the phylogenetic tree).

No further outbreak cases were observed in the two months after 6 December (i.e., after the onset date of the last outbreak case).

### 3.4. Risk Factor Analysis and Case Interview

Standard demographic and epidemiological information were collected for each case by standard questionnaires: upon admission for hospitalized cases, at the follow-up visit for transplanted people, and at the time of donation for blood donors. However, a further in depth interview of outbreak cases was also carried out to identify any possible common epidemiological links.

The results are summarized in Table 1. Forty-seven cases of HEV infection were observed between 8 June and 6 December. Gender information was available for 44 cases: the majority of them were males (34/44, 77.3%). The median age was 63 years (range: 28–85 years). Not surprisingly, the great majority of cases reported consumption of raw/undercooked meat and/or sausage and/or liver sausage from pigs or wild boar within two months before onset of symptoms. Table 1 also reports details in the patient subgroups (Cluster A, B, and C) identified by HEV phylogenetic analysis.

All other investigated risk factors (consumption of raw/undercooked meat from deer, consumption of shellfish, direct contact with pig or wild boar, small farming of pigs for self-consumption, exposure to pig manure, hunting, having drunk well water) were rarely or not at all reported (data not shown).

The identification of molecular clusters based on HEV sequencing permitted the targeted analysis of cases likely sharing a common source of infection, as shown by their identical or very similar strains. While the size of Cluster A was too small (4 cases, one of them with no other information than gender and age), the size of Cluster B (*n* = 15) and Cluster C (*n* = 11) was sufficient to draw some conclusions.

The patients in Cluster B included 9 cases from Lazio and 6 cases from Abruzzo. The cases from Lazio mainly included male subjects (7 M, 2 F) and the median age was 64 years (range: 46–85 years); importantly, seven cases reported a stay in Abruzzo in the province of L’Aquila and five cases referred to having consumed products of pig origin purchased or consumed in local businesses.

The six cases from Abruzzo also mainly included male subjects (4M, 2F) and median age was 57 years (range 46–77 years), which are not significantly different from the Lazio cases. Five of them resided in the province of L’Aquila; all of them reported having consumed products of pig origin and no stay/travel outside of Abruzzo. Overall, the data suggest a link with the consumption of products of swine origin in the province of L’Aquila.

Patients in Cluster C included seven cases from Lazio and four cases from Abruzzo. The patients from Lazio exclusively included male subjects (7 M, 0 F) and the median age was 69 years (range: 63–74 years). Four of them were residents in the province of Rieti, which is an area of Lazio bordering on the province of L’Aquila in Abruzzo. All seven cases reported consumption of products of swine origin (sausages and cured meats), five of them in locations in the province of Rieti or very close to Rieti but none reported any stay/travel in Abruzzo. The four cases from Abruzzo (3M, 1F) showed the median age of 51 years (range 38–60 years) and three of them were residents in the province of L’Aquila; importantly, all of them reported having consumed products of swine origin but none of them reported any stay/travel outside Abruzzo. Overall, the data suggests a link with the consumption of products of swine origin for Cluster C too; however, these cases appear more geographically widespread than cases from Cluster B, having occurred over a larger area of middle Italy including both the province of Rieti and the province of L’Aquila, i.e., including two geographically adjacent areas.

### 3.5. Food Investigation

As soon as the outbreak was recognized, the local food health services in Abruzzo were alerted. Careful investigation at restaurants/places linked to cases was carried out, including food sampling as well as tracing-back suspected food to distributors and producers. As a common denominator of such investigations, most lots/batches of suspected food were no longer available at the time of sampling. This was mainly due to the lag between consumption and the onset of symptoms, which led to delayed sampling. However, as an attempt, a different lot of the same food type was sampled whenever available. Search for HEV RNA gave negative results in all cases.

Several cases observed in Lazio were linked to a trip to Abruzzo or to residence in the province of Rieti (bordering on Abruzzo) and, in both cases, there was on-site consumption of pork products, which led to food investigation in those areas. In one case the suspected source of infection was unlinked to fall in Abruzzo or in the province of Rieti: the pork sausages were purchased at a supermarket near Rome. No sausage remnants were available at the patient’s home for laboratory investigation; the specific lot purchased by the case was no longer available at the supermarket and so a different lot was sampled that, however, tested HEV RNA negative.

### 3.6. Outbreak Control Measures

Sampling of the food batches suspected to be the source of infection was unsuccessful. This was largely a consequence of the lag between food consumption and onset of symptoms and, to a less extent, a consequence of the cumulative time needed for diagnosis, notification to food health services and food sampling. Even though this latter is amenable to slight improvements, the time between consumption and onset of symptoms, obviously, cannot be modified: thus, food investigation continues to be challenging and it is expected that it might be successful only in a very small fraction of cases.

The outbreak was self-limited: no more outbreak cases were detected within two months after 6 December 2019, onset date of the last outbreak case.

## 4. Discussion

All cases of the present outbreak were geographically linked to middle Italy areas in which HEV was previously shown to be hyper-endemic. In fact, a national retrospective study in blood donors showed very high anti-HEV IgG prevalence rates approaching or even higher than 30% in these areas in contrast with the 8.7% average national rate which confirms the findings of a previous study focused on a restricted area [19,31].

Overall, 47 cases of HEV infection were observed between June 2019 and December 2019. They represent a marked increase as compared with just a few cases in the same period in the past years in the same areas.

The phylogenetic analysis of viral sequences showed three different HEV clusters: two of them were caused by related yet distinct subtype 3e variants while the third clusterwas caused by subtype 3f viruses. The outbreak curve showed a partially overlapped temporal distribution of the three clusters. In the present study, the observed increase in cases was defined “an outbreak” according to the classical definition of epidemiology based on “cases of disease” (as usually used by WHO, ECDC, and CDC), i.e., “the occurrence of more cases of disease than expected in a given area during a specified period”.

However, the detection of three molecular clusters raises the question if a single outbreak sustained by three strains or, alternatively, three independent outbreaks, each one sustained by its own strain, was ongoing. The available data do not allow for a conclusive answer because no specific source(s) of infection could be identified: no investigated food showed detectable HEV (see Section 3.5). Analysis of epidemiological data could identify the consumption of pork products (sausage, liver sausage, and raw/undercooked meat) as the only factor shared by all but one cases, which suggests that pork products are the most likely source of infection. However, none of the specific products (sausage, liver sausage, and raw/undercooked meat) was common to a majority of cases. This is not surprising because infection can be transmitted by the consumption of any of the different food products derived from an infected animal, with the unique condition being the presence of a high enough dose of infectious virus. In addition, a recall bias can also affect the quality and quantity of information collected by interview due to the lag between food consumption and case interview. The 3e and the 3f outbreak strains appeared genetically unrelated to any HEV strains so far detected from the entire Abruzzo region. Moreover, in past years the 3e subtype had been observed exclusively in the coastal area of Abruzzo (Figure 1: ISS_ID259/2018 in Chieti and ISS_ID261/2018, ISS_ID60/2008, and ISS_ID145/2015 in Pescara) but not at all in the inner area involved in the present outbreak.

The cases caused by 3e and 3f outbreak strains showed the different extents of geographical distribution: the 15 cases of Cluster B (Figure 1, 3e outbreak strain) appeared restricted to the province of L’Aquila, while the 11 cases of Cluster C (Figure 1, 3f outbreak strains) occurred over a larger area in middle Italy, including both the province of L’Aquila in Abruzzo and the province of Rieti in Lazio. It is possible that these different distributions may be linked to the different commercial distributions of the involved pork products.

Overall, the findings suggest the outbreak might have been caused by newly and almost simultaneously introduced strains that were not previously circulating in this area and possibly harbored by pork products or live animals imported from outside Abruzzo. This possibility deserves further studies in this area to monitor the circulation of HEV in human cases as well as in pigs and wild boars. Prevention of future outbreaks will require coordinated efforts. Prevention might partly rely on information campaigns to give advice to the population that pork meat, sausages, and liver sausages must not be consumed raw or undercooked: this is a difficult task because some eating habits are based on long-standing traditions. In order to prevent future outbreaks, systematic sampling of pork products followed by testing for HEV RNA or in vitro infectivity are also potentially useful; however, both the sustainability from the organizational and economic point of view and its real impact on risk reduction has to be carefully evaluated. Furthermore, the need for standardized sensitive methods for the detection of HEV RNA or in vitro infectivity in food samples should be highlighted.

## Figures and Tables

**Figure 1 viruses-13-01159-f001:**
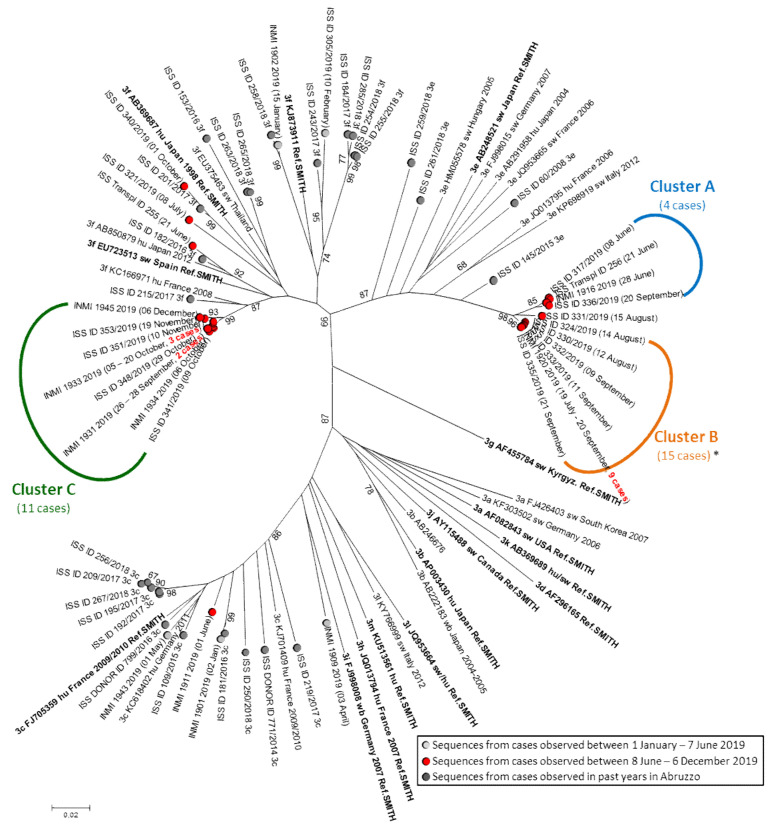
Phylogenetic tree of the HEV sequences. A red circle marks the HEV sequences obtained between 8 June and 6 December 2019 (including both sporadic and outbreak cases). Sequences from Abruzzo: the sequence label begins with “ISS”; sequences from Lazio: the sequence label begins with “INMI”. Some sequences are representative of a group of identical sequences (the overall number of represented cases is reported in bold red in the sequence label). A dark gray circle marks the HEV sequences obtained in Abruzzo in the years before 2019 and a light gray circle marks HEV sequences from cases with onset between 1 January and 7 June 2019. Reference genotype 3 strains with known subtype were included in the analysis (in bold: references proposed in [29] and/or [30]); subtype, accession number, host (hu: human; sw: swine, wb: wild boar; de: deer; mo: mongoose; ra: rabbit), and year of detection are reported for each reference sequence whenever available. Bootstrap values > 65 are shown. *: the 15 cases of Cluster B include a case for which a very short sequence could not be included in the phylogenetic analysis; however, it showed 100% identity with the sequences in Cluster B over the entire available length, 165 nt.

**Figure 2 viruses-13-01159-f002:**
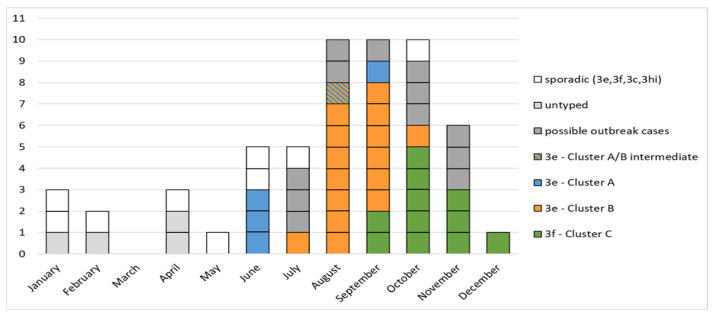
Outbreak curve by onset date. The grey boxes represent cases with no sequence available.

**Table 1 viruses-13-01159-t001:** Gender, age and risk factors of Cluster A, B, and C cases and 47 total cases of HEV infection with onset from 8 June to 6 December *.

	Cluster A (*n* = 4)	Cluster B (*n* = 15)	Cluster C (*n* = 11)	Total (*n* = 47)
Male/Female ratio	3/1	11/4	10/1	34/10
Male %	75%	73.3%	90.9%	77.3%
Median age (range)	59 (53–72)	61 (46–85)	66 (38–74)	63 (28–85)
Consumption of raw or undercooked pork meat	1/3 (33.3%)	8/13 (61.5%)	2/10 (20%)	15/39 (38.5%)
Consumption of pig sausage	3/3 (100%)	6/13 (46.1%)	8/10 (80%)	28/39 (71.8%)
Consumption of wild boar sausage	0/3 (0%)	2/12 (16.7%)	1/9 (11.1%)	7/37 (18.9%)
Consumption of liver sausage (pig or wild boar)	2/3 (66.7%)	7/13 (53.8%)	3/10 (30%)	17/39 (43.6%)
Consumption of any (raw/undercooked meat and/or sausage and/or liver sausage from pig or wild boar)	3/3 (100%)	13/13 (100%)	10/10 (100%)	38/39 (97.4%)

* For each variable, information was incomplete for some cases.

## Data Availability

The sequences of the present study were deposited in GenBank (Accession number MN497623, MN537879, MN737482, MN737483, MN737484, LN681545, and MZ274228 to MZ274271).

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
