# Peer review of "Hepatitis E Outbreak in the Central Part of Italy Sustained by Multiple HEV Genotype 3 Strains, June–December 2019"

_viruses, 2021, doi:10.3390/v13061159_

Round 1

Reviewer 1 Report

Garbuglia et al. described Hepatitis E outbreak in the central part of Italy mediated by  HEV genotype 3 strains mainly subtypes 3e and 3f,  during the period of June-December 2019. 47 cases of HEV infection were recorded, from them sequencing were successful in 35 cases. Phylogenetic analysis of the viral sequences showed 30 of them grouped in three distinct molecular clusters, termed A, B (subtype e) and C (subtype 3f). Pork products as the most likely source of the outbreak

Overall, the manuscript is interesting and of scientific merits especially for Italians 

I have some questions and/or suggestions

1) The medical data of the cases is poor, the authors should provide some data for 47 cases, or at least 35 sequenced cases including liver enzymes levels, bilirubin, hospitalization duration, age, presence of other diseases, ...etc.

For the transplanted patients, thw following data should be provided age, organ transplanted, treatment therapy, duration since organ transplantation, age,...etc

2) What is the status of anti-HEV IgG in these patients?

3)I  have a question regarding the Case definition, why the authors defined any viral sequences identical or clustering to Acc.No. MN497623, 95
MN537879, MN737483 as confirmed outbreak cases , while a viral sequence in the ORF2 region clustering in a phylogenetic tree in a different branch vs the following three HEV reference sequences: Acc.No. MN497623, MN537879, MN737483 as non outbreak cases. It could ne outbreak caused by mutliple subtypes, please verify.

4) Why authors only released few sequences to Genbank? I can understand due to the data privacy and novelty . But they can deposit the data into Genbank and select the data release after the paper processing and acceptance.

5) Can the authors provide level of HEV RNA load in the patients? Is there any difference in immunocompromised and immunocompetent regarding the viral load?

Reviewer 2 Report

Observational study about HEV; Nice presentation, but no data or clinical consequences about the usefulness of these informations: for example, are there data on the control of production, slaughtering, processing of pork in the province of Rieti or L'Aquila? Have the competent local authorities been notified of this epidemic, to intensify controls? Are there data or prospective studies, perhaps still in progress, regarding the disappearance of the factors that produced this epidemic? What kind of measures have been suggested to the various local authorities for follow up and prevention?

Reviewer 3 Report

In this manuscript (Viruses-1227866), authors described a HEV outbreak occurred in Central Italy. Most cases were linked to consumption of pork products at restaurants or festivals in Abruzzo Region; the outbreak strains formed three clusters independent from the strains detected in previous years in sporadic cases from this area.

Comments:

(1)  Pages 1-2, Introduction: all taxonomic ranges have to be written in italics. For example: Hepeviridae, Orthohepevirus.

(2)  Page 2, Materials and Methods: authors should expand this section. They should indicate the test used for the detection of HEV RNA and anti-HEV-IgM. What sensitivity and specificity did these tests have?.It would also be interesting if the biochemical levels of these patients were indicated (transaminases, bilirrubin…). 

(3)  Page 7. Results: only one patient with HEV infection had not consumed raw/undercooked meat or sausage or liver sausage from pig or wild board. Do the authors know which epidemiological risk factor was associated with this infection? 

(4)  Table 1. It could be divided according to the clusters and put a column with the total. 

(5)  Discussion should be expanded. What would be recommended so that these outbreaks do not occur?.

Reviewer 4 Report

The authors describe an outbreak of hepatitis E caused by HEV genotype 3 from two Italian regions, Abruzzo and Lazio. Overall, 47 cases of HEV infection were reported in the second half of 2019, with an apparent increase on a year-to-year basis. Partial viral sequences were determined for 35 cases and they were found to fall into three distinct genetic clusters (A, B and C). Epidemiological data identified pork products as the possible source. The Authors conclude that the pork products or live animals imported from other geographical areas could be the origin of the outbreak.

Overall, the MS seems well written but I have some concerns/suggestions.

i) The accessions of the sequences should be provided, as they were not included in the MS. This is unusual. Also, in the Materials and methods I would appreciate a clearer description of the RT-PCR assays used in the study.

ii) Representatives of the three main clusters, at least 1 strain for cluster, could be selected for full genome sequencing, maybe samples with the highest viral load. This would add depth to the study.

iii) The authors claim that an outbreak, defined as an excess of cases on a year-to-year basis, is actually sustained by different clusters/lineages. Is correct to define this picture as a single outbreak if there are cases from different geographical areas, apparently epidemiologically unrelated? 

Consumption of pork meat is recognized as risk factor, but is this enough to define a common origin? Are there other epidemiological data available from the patients?

iv) The English is generally good but it is odd in some points and should be revised carefully.

Round 2

Reviewer 1 Report

I checked the manuscript and replies of the authors to my comments, the authors have replied adequately to my questions and the manuscript has been improved significantly. I want to thank the authors for doing this.

I have a minor suggestion in the introduction section.

Page 2 lines 65-68: : HEV-3 and HEV-4 have been detected both in humans and in animal species (HEV-3 in pig, wild boar, deer, mongoose, rabbit; HEV-4 in pig). HEV-3 and HEV-4 are mainly transmitted by consumption of products from infected animals, especially undercooked meat (including pork liver and liver containing pork products) [8]

Here, I suggest the authors to include the other animals that recently recorded as a potential HEV reservoir such as cow, sheep, and goat. Animal products such as liver and milk could be reported as source of HEV infection. This finding is reported in Europe, China, Egypt, and Turkey. 

PMID: 33322702, PMID: 32659521, PMID: 30683038 , PMID: 31874303, PMID: 27286751.

Also HEV was recorded in sheep and goat in Italy: PMID: 30811818, PMID: 27647265.

Therefore, ruminants and their products could be directly on indirectly cause of infection/outbreaks in Italy.

Please include this information and cite the above mentioned references.

2- page 2 line 60: HEV-1 and HEV-2 infect exclusively humans, show fecal-oral transmission through contaminated water and are responsible for both sporadic cases and waterborne epidemics.

Please add that HEV-1 could to acute liver failure especially in old age and leukemic patients PMID: 33469320 , PMID: 34002677.

Congratulations for this nice work

Author Response

Comments and Suggestions for Authors

I checked the manuscript and replies of the authors to my comments, the authors have replied adequately to my questions and the manuscript has been improved significantly. I want to thank the authors for doing this.

I have a minor suggestion in the introduction section.

Point 1: Page 2 lines 65-68: : HEV-3 and HEV-4 have been detected both in humans and in animal species (HEV-3 in pig, wild boar, deer, mongoose, rabbit; HEV-4 in pig). HEV-3 and HEV-4 are mainly transmitted by consumption of products from infected animals, especially undercooked meat (including pork liver and liver containing pork products) [8]

Here, I suggest the authors to include the other animals that recently recorded as a potential HEV reservoir such as cow, sheep, and goat. Animal products such as liver and milk could be reported as source of HEV infection. This finding is reported in Europe, China, Egypt, and Turkey. 

PMID: 33322702, PMID: 32659521, PMID: 30683038 , PMID: 31874303, PMID: 27286751.

Also HEV was recorded in sheep and goat in Italy: PMID: 30811818, PMID: 27647265.

Therefore, ruminants and their products could be directly on indirectly cause of infection/outbreaks in Italy.

Please include this information and cite the above mentioned references.

Response 1: All the references were included in the Introduction section, as requested

Point 2: page 2 line 60: HEV-1 and HEV-2 infect exclusively humans, show fecal-oral transmission through contaminated water and are responsible for both sporadic cases and waterborne epidemics.

Please add that HEV-1 could to acute liver failure especially in old age and leukemic patients PMID: 33469320 , PMID: 34002677.

Congratulations for this nice work

Response 2: The two references were included in the Introduction section, as requested

Reviewer 4 Report

The authors have addressed/replied to my specific comments.

1) Accessions were provided and details of the Methods were added. This point is resolved

2) While reasoning on the partial sequencing strategy is acceptable in strict terms of epidemiology, gathering genome sequence data would be beneficial for the MS, at least for overrappresented strains, in terms of virology. Since a quantitiative RT-PCR was used for diagnosis, it could be easy to pick up HEV strains with high viral load, if any, for genome sequencing. This information (quantitation of HEV in patients) was not provided in the MS, if I am correct. Were there differences in terms of virus titre in the patients among the three HEV clusters? Genome recombination is common in ssRNA + viruses and this relevant information will be missing. Yet I understand that this would delay significantly publication of the MS.

3) Since there are at least 3 different HEV strains implied in the "outbreak", I do not agree that this is a unique outbreak. Even if the majority of patients had consumed pork meat, this is not enough to consider this as a unique event. Why the Authors rule out that multiple independent outbreaks were ongoing? The Authors could comment on this in the discussion.

4) The English is good but still odd in some points
